# Burnout and Depression in Portuguese Healthcare Workers during the COVID-19 Pandemic—The Mediating Role of Psychological Resilience

**DOI:** 10.3390/ijerph18020636

**Published:** 2021-01-13

**Authors:** Carla Serrão, Ivone Duarte, Luísa Castro, Andreia Teixeira

**Affiliations:** 1School of Education-Polytechnic of Porto, Portugal and Centre for Research and Innovation in Education, 4200-465 Porto, Portugal; 2Faculty of Medicine, MEDCIDS-Department of Community Medicine, Information and Decision in Health, University of Porto, 4200-450 Porto, Portugal; luisacastro@med.up.pt (L.C.); andreiasofiat@med.up.pt (A.T.); 3Faculty of Medicine, CINTESIS-Center for Health Technology and Services Research, 4200-450 Porto, Portugal; 4School of Health Sciences-Polytechnic of Porto, 4200-450 Porto, Portugal; 5IPVC–Instituto Politécnico de Viana do Castelo, 4900-347 Viana do Castelo, Portugal

**Keywords:** COVID-19, healthcare workers, resilience, depression, burnout

## Abstract

During the COVID-19 pandemic, healthcare workers (HCW) have been exposed to multiple psychosocial stressors. Resilience might protect employees from the negative consequences of chronic stress. The aim of this study was to explore the mediating role of resilience in the relationship between depression and burnout (personal, work-related, and client-related). A cross-sectional study was performed using an online questionnaire distributed via social networks. A survey was conducted comprising standardized measures of resilience (Resilience Scale-25 items), depression (subscale of Depression Anxiety Stress Scales-21 items), and burnout (Copenhagen Burnout Inventory Scale-19 items). A total of 2008 subjects completed the survey, and a hierarchical regression model was estimated for each burnout dimension. The results revealed that depression had not only a directed effect on personal, work- and client-related burnout, but also an indirect small effect on it through resilience. Psychological resilience played a partial mediating role between depression and all burnout dimensions. This partial mediation suggests that there may be other possible variables (e.g., social connection, self-compassion, gratitude, sense of purpose) that further explain the associations.

## 1. Introduction

The COVID-19 pandemic is a global public health emergency with tremendous consequences for people’s lives and their mental health. On 2 March 2020, Portugal announced the first confirmed case of SARS-CoV-2. On 9 May 2020, 26,715 cases of COVID-19 were recorded. One month later, the number of cases was confirmed as 35,910 with more than 1500 deaths [1]. Healthcare workers (HCW) had to deal with equal consequences as the general population, in addition to having to respond to substantial challenges. HCW are at increased risk of acquiring and potentially transmitting COVID-19 to patients, coworkers, and family, have to work with new and constantly changing protocols, have to deal with a heavier and more stressful workload, limited personal protective equipment, care for critically ill patients [2,3,4]. If prolonged stress and the symptomatology associated with working conditions to which HCW were exposed were already a concern before the pandemic e.g., [5,6], the pandemic may have exacerbated this psychosocial vulnerability. Two Portuguese studies have reported a high prevalence of burnout among health professionals. In one of these studies [5], a nationwide study of burnout, among 1262 nurses and 466 physicians, authors reported that 21.6% of HCW showed moderate burnout and 47.8% showed high burnout. Similar high risk of burnout was found in both professions, with no significant differences between the two groups. In the study by Reis [6], carried out in primary care units, 66% of physicians had a high emotional exhaustion score, 45.7% had a high depersonalization score, and 48.2% of physicians had low scores on personal accomplishment at work.

Burnout and psychological distress seem to have been some of the immediate effects of the pandemic on health professionals [3,4,7,8]. Burnout is defined as a “state of physical, emotional and mental exhaustion that results from long-term involvement in work situations that are emotionally demanding” [9]. Kristensen et al. [10] indicate that the core of burnout is fatigue and exhaustion, in three contexts, i.e., person (personal/self) related, client-related, and work/workplace. Psychological distress is “a state of emotional suffering characterized by symptoms of depression and anxiety sometimes accompanied by somatic symptoms, several characteristic features emerge” [11]. 

In this pandemic scenario, fear of COVID-19 contagion seems to be an exacerbator of distress and has been found to be associated with negative mental health outcomes [4,12,13]. In this line, Wankowicz et al. [14] reported that healthcare workers who are exposed to SARS-CoV-2-infected patients in emergency wards, infectious wards, and intensive care units are at a much higher risk of experiencing symptoms of anxiety and depression than HCW working in other wards.

A cross-sectional web survey [3] of New York physicians (n = 657), concluded that three out four HCW were highly distressed by fears about transmitting COVID-19 to family or friends, and 48% screened positive for depression.

According to Luceño-Moreno et al. [13], in Spain, more than half had posttraumatic stress disorder and anxiety disorder and almost half of the sample had depressive disorder.

Another study developed in China shows that about half (50.4%) of the HCW reported symptoms of depression [8].

Weilenmann et al. [15], in Switzerland, explored the level of burnout and psychological distress (depression and anxiety) in 1410 HCW. The results showed a high level of burnout and that 25.9% and 20.7% had clinical levels of anxiety and depression, respectively. 

Burnout can affect health, leading to the development of physical and psychosomatic symptomatology and depression [16]. Burnout can reduce the quality of care [17], cause mistakes in the healthcare provided, lead to higher occurrence of medical leave and absenteeism [5,16].

It is also worth noting that not all individuals exposed to very intense crisis situations suffer from mood disorders, with resilience being vital as a protective variable [18,19,20,21] and as a burnout protection factor [13,20,22,23].

### Psychological Resilience as a Factor for Protecting from Burnout and Depression

The existing literature considers that psychological resilience represents an adaptive quality that allows the person to grow in consequence of experiencing traumatic situations and high stress [24,25]. Studies have suggested that resilience is synonymous with reduced vulnerability [26], with the ability to adapt to adversity. Resiliency is an adaptive personal resource against significant stressors of all kinds, including work stressors [20,24]. As adapted to the workplace, resiliency has been defined as the “positive psychological capacity to rebound, to ‘bounce back’ from adversity, uncertainty, conflict, failure, or even positive change, progress and increased responsibility” [27]. Resilience minimizes and protects against negative, stress-related effects, such as burnout syndrome [21,23,28]. 

West et al. [29] examined a sample of 5445 US physicians and concluded that resilience was inversely associated with burnout symptoms, but burnout rates were substantial even amongst the most resilient physicians. 

Resilience seems to have a mediation role in some psychological variables [30]. Little research has considered how resilience, and burnout and psychological distress are related and interact. For example, in an intensive care setting, Arrogante and Aparicio-Zaldivar [31] concluded that resilience mediated the relationships between emotional exhaustion and depersonalization (Maslach Burnout Inventory; [16]) with mental health (partial mediations) and the relationship between personal accomplishment and mental health (total mediation). In a study involving 194 nurses from Madrid (Spain), the authors showed the relationship between burnout dimensions, physical and mental health and resilience, and analyzed the mediating role of resilience in the relationship between burnout (Maslach Burnout Inventory; [16]) and health. The results showed that mental health was negatively related with the three dimensions (emotional exhaustion, depersonalization, and personal accomplishment) of burnout and positively related with resilience. He concluded that resilience is not only relevant to improve the mental health of nurses, but also to amortize and minimize the negative consequences of work stress and whose most adverse outcome is burnout [32]. However, it is unclear whether resilience plays a mediating or moderating role [33]. In addition, two other questions currently remain open: (1) is it burnout that triggers depression or vice versa [34]; (2) depression and burnout do or do not overlap with each other [35,36]. 

Thus, understanding resilience is important as a means to developing interventions to prevent/treat mental disorders and promote mental health. The issue of resilience and the possible benefits of resilience training are particularly relevant in professional groups who are exposed to various sources of stress [25,37]. 

Considering the available evidence that psychological resilience might protect workers from the negative consequences of chronic stress e.g., [3,25,37], the aim of this study was to explore the mediating role of resilience in the relationship between depression and burnout (personal, work-related, and client-related).

## 2. Materials and Methods 

### 2.1. Study Design and Participants

This was a cross-sectional quantitative study directed to HCW living in Portugal. Data was collected through a web survey disseminated on social networks using and spread through the snowball sampling technique. The data collection period spanned from 9 May to 8 June 2020. In Portugal, during this period, national calamity was declared, and the state of national emergency was followed by a relieving of lockdown measures (between 18 March and 2 May). A questionnaire was made available to participants via a link which was spread by professional organizations (e.g., National Union of Senior Health Technicians in the Diagnostic and Therapeutic Areas, Portuguese Nurses Association, Portuguese Association of Physiotherapists, Order of Nutritionists, Order of Portuguese Nurses, Portuguese Nutrition Association, Nutritionists Portugal, Portuguese Nurses Union, Independent Doctors Union, Order of Portuguese Psychologists and the Regional Pharmacovigilance Units) and healthcare institutions by e-mail and social networks. The questions were answered online through the Google^®^ Forms platform. Subjects who did not develop their professional activity in the health field and who answered all items of the Scales were excluded. As inclusion criteria, all subjects working in Portugal during the data collection period were included. Ethical procedures according to the Declaration of Helsinki were accomplished via analysis and approval of the study by an independent Ethical Committee (Ethics Committee of São João Hospital Center) (Ref 184/2020 on 7 May 2020).

A total of 2008 HCW completed the questionnaire. There was a marked predominance of females (1678, 83.6%). This distribution of participants may be due to the high level of feminization of health professionals in Portugal. The global feminization rate in Health is 76% [38]. The average age was 38 years old (SD = 10 years old). The most common marital status was married or living as a couple (53.3%) followed by being single (38.8%). Most of the participants were graduates (60.1%) and 39.9% were postgraduates.

The participants of this study were 707 (35.2%) allied health professionals (defined as healthcare professions distinct from dentistry, nursing, medicine, and pharmacy and who provide a range of diagnostic, technical, therapeutic, and support services in connection with healthcare), 511 (25.4%) physicians, 409 (20.4%) nurses, 88 (4.4%) pharmacists, 83 (4.1%) psychologists, 72 (3.6%) nutritionists, 29 (1.4%) healthcare assistants, and 21 (1%) workers in allied areas. Of all of the participants, 485 (24.2%) were in primary healthcare, 383 (19.1%) in inpatient areas, 247 (12.3%) in emergency services, 167 (8.3%) were in COVID-19 inpatient areas exclusively, and 157 (7.8%) worked in high-dependency units (intermediate and intensive care). Amongst the participants, 524 (26.1%) revealed having health problems such as chronic respiratory disease (158, 30.2%) and a compromised immune system (119, 22.7%). A total of 319 (15.9%) subjects were caring for older people or with disabilities. In Table 1, the characteristics of the participants are synthetized.

### 2.2. Survey Questionnaire

The questionnaire comprised two parts: epidemiology (age, sex, marital status, parental status, years of professional experience, salary reduction, previous medical history, frontline, etc.), psychological resilience, psychological distress (depression, anxiety, and stress), and burnout. The Resilience Scale [30] was used to measure psychological resilience. This scale comprises 25 items with a seven-point Likert response scale ordered from strongly disagree (one) to strongly agree (seven). The Portuguese version of Resilience Scale presented high internal consistency, with a Cronbach’s alpha (α) of 0.89 [39]. In the current study, a Cronbach’s alpha of 0.95 was obtained. The Depression, Anxiety and Stress Scale (DASS-21; [40]), Portuguese version [41] was used to assess depression state among respondents. The full scale contains three subscales measuring depression, anxiety, and stress. The depression subscale contains seven items, and the respondents are asked to rate the extent to which they have experienced each state over the past week using a scale of zero (did not apply to me at all), one (applied to me to some degree, or some of the time), two (applied to me to a considerable degree, or a good part of time), and three (applied to me very much or most of the time). In this study, the Cronbach’s alpha was 0.90. Burnout was measured by the Copenhagen Burnout Inventory (CBI; [10]) which is an instrument (19 items) with three subscales: six items for personal burnout, seven for work-related burnout, and six items for client-related burnout. The subscale of personal burnout measures feelings of emotional, physical, and mental exhaustion and fatigue. Symptoms that respondents’ attribute to work are assessed by the work-related burnout subscale. The subscale of client-related accounts for feelings of psychological and physical fatigue and exhaustion that respondents allocate to their work with clients or patients, in this case. Items are answered on a five-point Likert scale response (always/to a very high degree = 100, often/to a high degree = 75, sometimes/somewhat = 50, seldom/to a low degree = 25, and never /almost never/to a very low degree = 0). For each subscale, the score corresponds to the average scores of items within that subscale and range from 0 to 100. High-level burnout was considered when scores of 50 or above were obtained in each of the subscales [10,42]. High internal consistency has characterized the three subscales, in the original version [10] (α = 0.84) and in the Portuguese version [42] (α = 0.86). In the present study, Cronbach’s alphas for personal, work-related, and client-related burnout were 0.91, 0.89, and 0.89, respectively.

### 2.3. Statistical Analysis

Data analysis was performed using SPSS^®^ Statistics (version 26, IBM, Armonk, NY, USA) and Jamovi software (datalab.CC, Sydney, Australia). Categorical variables were described by absolute and relative frequencies. Quantitative normally distributed variables were described by the mean and the respective standard deviation (SD). Quantitative non-normally distributed variables were described by the median (Mdn) and the respective interquartile interval (Q1; Q3). The normality of distributions was verified by observation of the respective histograms. For each outcome (personal burnout, work-related burnout, and client-related burnout), a separated multiple linear regression was performed. To decide which independent variables to include in each multiple regression, simple linear regressions were performed with each of the following variables: sex, marital status (married; single; divorced or separate; widowed), children (≤12 years old; no children or > 12 years old), educational level (High school and below; University degree; Postgraduate; Master’s; PhD), co-living (friends; family; alone), caregiver (no/yes), lives with a person at risk for COVID-19 infection (no/yes), death of relative or friend during the pandemic period (no/yes), years of professional experience (≤5 years; 6–15 years; >15 years), salary reduction (no/yes), frontline working position (no/yes), diagnosed health problem (no/yes), COVID-19 tested (yes; no, but I’d like to do it; no, I have no interest) and direct contact with infected people (no/yes). All variables that correlated with the outcomes at *p* ≤ 0.20 in a simple regression were included in the multiple linear regressions. Only the significant variables were maintained in the final multiple models. A hierarchical regression model was estimated to examine the mediating role of resilience in the relationship between depression and each burnout dimension: personal burnout, work-related burnout, and client-related burnout. The following requirements for such analysis were verified: a significant correlation between depression (independent variable) and burnout (dependent variable); a significant correlation between depression and resilience (the mediator) and between resilience and burnout. Additionally, the effect of depression on burnout should shrink (partial mediator) or become statistically insignificant (full mediator) after the inclusion of resilience in the model. Standardized estimates (β), F statistics, determination coefficient (R^2^), and R^2^-changes (ऴR^2^) for each step were provided. Multicollinearity was checked through tolerances and variance inflation factors ranges. Finally, the Sobel test was pursued to assess the mediation effect.

Values of *p* ≤ 0.05 were considered significant. 

## 3. Results

In general, participants showed a moderate level (50.8%) or high level of psychological resilience (27.8%), normal levels for depression (70.6%), high levels of work, personal and client-related burnout, 53.1%, 52.5%, 35.4%, respectively. 

### 3.1. The Mediating Role of Resilience in the Relationship between Depression and Personal Burnout

The mediating role of resilience in the relationship between depression and personal burnout (outcome) was explored through hierarchical linear regression analyses and the results are demonstrated in Table 2. The model comprises three steps: in the first step, all of the independent variables considered associated with personal burnout were adjusted in a multiple linear regression (sex, marital status, parental status, frontline working position, diagnosed health problem, COVID-19 tested, and direct contact with infected people); in the following two steps, depression was entered and then resilience was included. The obtained model showed a positive association between personal burnout and depression, explaining 27.2% of personal burnout data variance (β = 0.530, *p* < 0.001), and a negative association between personal burnout and resilience, accounting for an increase of 1.5% in the explained variance (β = −0.132, *p* < 0.001).

Given that the absolute value of the depression’s standardized regression coefficient (β) reduced from 0.530 to 0.480 after the inclusion of resilience in the model (Sobel test, z = 6.47, *p* < 0.001), resilience was found to play a partial mediating role in the association between depression and personal burnout (Figure 1). Multicollinearity was not problematic since tolerance range was 0.864–0.994 and variance inflation factors varied between 1.01 and 1.16.

### 3.2. The Mediating Role of Resilience in the Relationship between Depression and Work-Related Burnout

The mediating role of resilience in the relationship between depression and work-related burnout (outcome) was explored through hierarchical linear regression analyses and the results are demonstrated in Table 3. After adjusting to all of the independent variables considered associated with work-related burnout in a multiple linear regression (sex, parental status, educational level, professional experience, frontline working position, diagnosed health problem, COVID-19 tested, and direct contact with infected people) in step 1, depression was included in step 2 and resilience was inserted in the model in step 3. The obtained model showed a positive association between work-related burnout and depression, explaining 26.3% of burnout variance (β = 0.522, *p* < 0.001), and a negative association between work-related burnout and resilience, accounting for an increase of 1.2% on the explained variance (β = −0.118, *p* < 0.001). 

Resilience was found to play a partial mediating role in the association between depression and work-related burnout (Figure 2) since β has reduced from 0.522 to 0.476 (Sobel test, z = 5.661, *p* < 0.001). Multicollinearity was not problematic since tolerance range was 0.909–0.992 and variance inflation factors varied between 1.01 and 1.10.

### 3.3. The Mediating Role of Resilience in the Relationship between Depression and Client-Related Burnout

The results from the hierarchical linear regression model for exploring the mediating role of resilience in the relationship between depression and client-related burnout (outcome) are demonstrated in Table 4. In the first step of the model, all of the independent variables considered associated with client-related burnout were adjusted in a multiple linear regression (professional experience, direct contact with infected people, and death of relative or friend during the pandemic period). Then depression was inserted in step 2, and in step 3 resilience was included in the model. The results revealed a positive association between depression and client-related burnout, explaining 12.2% of its variance (β = 0.352, *p* < 0.001), and a negative association between resilience and client-related burnout, accounting for an increase of 1.2% in the explained burnout variance (β = −0.120, *p* < 0.001). 

Resilience was found to play a partial mediating role in the association between depression and client-related burnout (Figure 3) since β has dropped from 0.352 to 0.305 (Sobel test, z = 5.146, *p* < 0.001). Once again, multicollinearity was not problematic since tolerance range was 0.918–0.998 and variance inflation factors varied between 1.00 and 1.09.

## 4. Discussion

The aim of the present study was to analyze the potential mediating role of psychological resilience in the impact of depression on burnout among Portuguese HCW. Psychological resilience means that individuals can adapt and respond to difficulties, traumas, tragedies, and adversity [24]. In this sense, when individuals experience multiple stress and adverse circumstances, such as the COVID-19 pandemic, resilience is the adaptation process against distressing events, and can be an effective factor in maintaining mental health [25,37]. In pandemic situations, psychological distress seems to have an immediate effect on health professionals [3,4,7,8]. Our finding that 29.4% of the investigated HCW had clinically relevant symptoms of depression, was significantly higher than in another study conducted in Singapore who investigated a sample of 470 HCW. In this study the authors used the same scale (DASS-21) and the results demonstrated that 8.9% of participants screened positive for depression, during the COVID-19 pandemic [43]. Indeed, in our study depression scores were significantly higher in HWC, what can be related to the fact that Singapore’s study was performed earlier in the outbreak. Additionally, Singapore was affected by the SARS pandemic at the beginning of the century. Thus, probably these HCW had more experience dealing with a pandemic than Portugal’s HCW, which could have reduced their symptom burden [43]. In other study in Switzerland with 1410 physicians and nurses demonstrated that 20.7% had clinically relevant symptoms of depression [15], but lower than those found in a study conducted in Wuhan (the epicenter of the outbreak in China) [44] indicating that 58% had symptoms related to depression (958 HCW). Another study [45] showed that 45% of a sample of 2014 nurses with direct contact with infected people experienced depression (measured by Zung’s Self-Rating Depression Scale) with 14% having moderate to severe depression. This study showed that nurses with direct contact with infected people suffered from fears of infection and death [45]. Additionally, Wankowicz et al. [14] reported that healthcare workers who are exposed to SARS-V-2-infected patients are at a much higher risk of experiencing symptoms of anxiety and depression than HCW working in other wards.

The results show that more than half of HCW presented high levels of fatigue and exhaustion related to work (53.1%), personal (52.5%), and client-related burnout (35.4%). These results are substantially higher than those found in the study developed in India with a sample of 2026 HCW [46], where the prevalence of personal burnout was 44.6% (903) and work-related burnout was 26.9% (544). Compared to the previous report [5], in the period pre-COVID-19, on 1728 Portuguese HCW, 21.6% showed moderate burnout and 47.8% showed high burnout, the incidence burnout in our study was relatively higher. It is not unexpected that SARS-CoV-2 has posed unprecedented challenges to HCW. Previous research on burnout has already found that the highest prevalence rate of burnout occurs among HCW in hospital emergencies [14]. Thus, in a pandemic, exacerbation of this situation would be expected. However, it is difficult to compare with previous literature as most studies have used different scales. This data is worrisome since the burnout can reduce the quality of care [17] and create mistakes in healthcare [5,16]. 

Our data suggests that HCW have a moderate or high level of psychological resilience, which echoes recent findings in the literature [13,45,47]. Although most individuals are resilient and do not develop clinical responses to trauma, they can still experience subclinical symptoms which have repercussions on personal dimension, as well as on professional performance [2]. Furthermore, the associations between psychological resilience, burnout, and depression, were also consistent with existing literature [13,45]. As expected, psychological resilience was negatively and significantly associated with depression and burnout. In other words, those higher in resilience experienced lower levels of depression and burnout. One study of 2014 frontline nurses from two hospitals in Wuhan showed that burnout, anxiety, and depression were moderately negatively correlated with resilience [45]. 

There is little research which has investigated the mediating role of resilience between depression and burnout. Moreover, in recent times the discussion has focused on whether depression and burnout are equal [36] or different constructs [35]. Koutsimani, Montgomery, and Georganta [35] researched the relation and distinction between burnout and depression by conducting systematic and meta-analysis review. The conclusion of the review was “although burnout and depression are associated with each other, the effect size is not so strong that it would suggest they are the same construct” (p.14). Finally, some authors suggest that it is depression that causes burnout and others that it is burnout that causes depression [34,35,36]. In fact, the underlying psychological mechanisms are not well understood. In addition, previous studies showed that symptoms of depression are associated with resilience [48] and high scores of psychological resilience corresponding to lower symptoms of depression [49]. As such, in this study we hypothesized that resilience could mediate the association between depression and burnout.

The mediational models obtained show that resilience seems to partially mediate the relationships amongst depression and all dimensions of burnout (personal, work-, and client-related burnout). Depression had not only a directed effect on personal, work-, and client-related burnout, but also an indirect small effect on it through resilience. It should be noted that this partial mediating effect may have been detected by the large sample size, since it produces a small increase in the determination coefficient. The HCW who scored higher on depression had lower resilience, resulting in higher levels of three dimensions of burnout, while those who scored low on depression had higher resilience, contributing to lower levels of personal, client-, and work-related burnout. The partial mediation suggests that there may be other possible variables (e.g., social connection, self-compassion, gratitude, sense of purpose; [2]) which further explain the associations. Despite it being a partial mediation, the findings suggest that for those who feel dysphoria, discouragement, lack of interest or involvement, anhedonia, and inertia, the resilience person’s capacity may help to reduce the impact of depression on burnout. While it would be crucial to carry out a detailed assessment to determine a clinical diagnosis.

Burnout is a response to stressful events [9,10,28] and how each individual responds to such events depends on how he/she evaluates them [10]. It is important that professionals are aware of the natural effect of these stressors [2] and their individual impact, and shame and guilt should not constitute avoidance behavior in the call for psychological support [4].

In burnout, it should be noted that over half of these professionals have high scores in work-related and personal burnout, but low in client-related burnout. Despite the fatigue and exhaustion, they felt inherent in overwork and multiple other stressors, with direct repercussions on personal and family life in which, their supportive and caring performance was not perceived in the same way. This result may be due to society’s broad recognition of health professionals, their perception of value, and self-recognition. On the other hand, training may have helped to achieve this result. Recent research has indicated that, during the COVID-19 outbreak, a significant percentage of HCW have high levels of emotional exhaustion, low in depersonalization and very high in personal accomplishment [13]. 

Exposure to COVID-19 patients had significant effects while the independent variable on all burnout dimensions. This result indicates that HCW who were in contact with infected patients were more likely to experience personal burnout, work-related burnout, and client-related burnout. This result highlights the need for organizations to develop protective strategies [14,21,22] in order to support these professionals in dealing with emergency situations. Attention to self-care needs is foundational for effective coping and cognitive functioning [2]. All of these findings in the present study contributed to the understanding of the relationship between resilience, depression, and burnout, and verified the fact that resilience is a tool for combating burnout among HCW [13,21,23].

Our study had the following limitations. Firstly, the study is based on a web-based survey, disseminated through email and social networks, which might have been affected by self-selection bias. For example, we can hypothesize that HCW who were most likely to respond to the survey were those with digital literacy or those who were more aware of the problems caused by burnout. Secondly, this study was a cross-sectional investigation, so interpretation of the results of mediation results on cross-sectional data must proceed with caution. Thirdly, the study was carried out during a specific pandemic period, meaning that it is necessary to employ a longitudinal design which examines the long-term effects of the pandemic in HCW and the level of psychological resilience. Finally, this study suggests that there may be other potential variables which elucidate the associations between depression and burnout. In this sense, it would be useful to consider other internal and external variables in future research. 

## 5. Conclusions

HCW living in Portugal experienced a high prevalence of burnout. After adjusting independent variables (e.g., sociodemographic and context variables), depression was positively associated with burnout dimensions and negatively associated with resilience. Furthermore, resilience could partially mediate the relationship between depression and dimensions of burnout. Considering that a second wave of infections is already happening and that there is a rise in workload in a context of doubt and insecurity, it is expected that burnout and depression might worsen. The solution will require increased funding for mental health, particularly for professionals who report symptoms of psychological distress (depression, anxiety).

## Figures and Tables

**Figure 1 ijerph-18-00636-f001:**
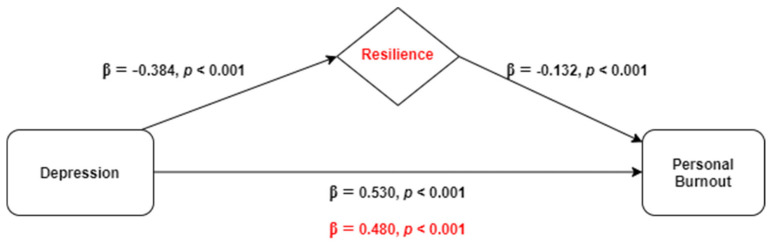
Representative scheme of the mediating role of resilience in the relationship between depression and personal burnout. Changes in beta weights when the mediator is present are highlighted in red.

**Figure 2 ijerph-18-00636-f002:**
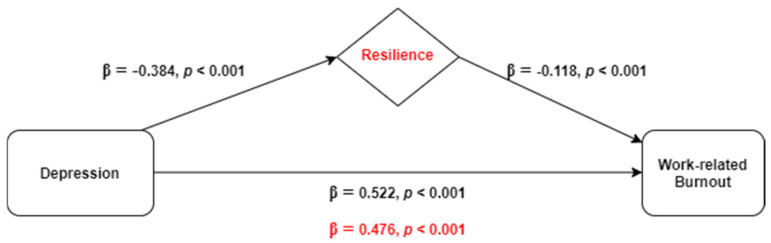
Representative scheme of the mediating role of resilience in the relationship between depression and work-related burnout. Changes in beta weights when the mediator is present are highlighted in red.

**Figure 3 ijerph-18-00636-f003:**
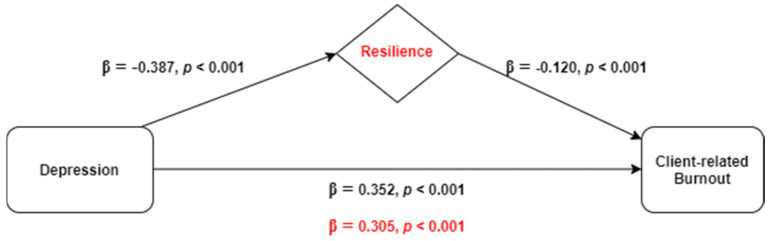
Representative scheme of the mediating role of resilience in the relationship between depression and client-related burnout. Changes in beta weights when the mediator is present are highlighted in red.

**Table 1 ijerph-18-00636-t001:** Characteristics of participants (n = 2008).

Characteristics	n (%)
Sex	
*Female*	1678 (83.6)
*Male*	330 (16.4)
Parents	
*Yes*	975 (48.6)
*No*	1033 (51.4)
Lives with a person at risk for COVID-19 infection	
*Yes*	681 (33.9)
*No*	1327 (66.1)
Death of relative or friend during the pandemic period	
*Yes*	118 (5.9)
*No*	1890 (94.1)
Professional experience	
*Five years or less*	504 (25.1)
*From 6 years to 15 years*	745 (37.1)
*More than 15 years*	759 (37.8)
Frontline working position ^a^	
*Yes*	1398 (69.7)
*No*	609 (30.3)
Direct contact with infected people	
*Yes*	552 (27.5)
*No*	1456 (72.5)
Salary reduction	
*Yes*	710 (35.4)
*No*	1298 (64.6)
Diagnosed health problem	
*Yes*	524 (26.1)
*No*	1484 (73.9)
COVID-19 Tested	
*Yes*	504 (25.1)
*No, but I’d like to do it*	983 (49.0)
*No, I have no interest*	521 (25.9)
	Mean (SD)
Personal burnout	49.3 (20.7)
Work-related burnout	50.2 (19.3)
Client-related burnout	38.9 (22.6)
	Mdn (Q1; Q3)
Resilience	137 (123; 146)
Depression	2 (1; 5)

^a^ Frontline HCW were defined as those who indicated they worked face to face, full time, or part-time.

**Table 2 ijerph-18-00636-t002:** Hierarchical linear regression analysis results (outcome variable: personal burnout).

Variables	Step 1 (β)	Step 2 (β)	Step 3 (β)
Sex	0.382 **	0.291 **	0.283 **
Marital status			
*Married/nonmarital partnership*	*Reference*		
*Single*	−0.018	−0.104 *	−0.107 *
*Divorced or separated*	0.012	−0.046	−0.057
*Widowed*	−0.228	−0.222	−0.238
Children ≤12y	0.196 **	0.214 **	0.208 **
Frontline working position	0.243 **	0.252 **	0.242 **
Diagnosed health problem	0.303 **	0.161 **	0.156 **
COVID-19 tested			
*Yes*	*Reference*		
*No, but I’d like to do it*	0.031	0.013	0.015
*No, I have no interest*	−0.216 **	−0.168 **	−0.169 **
Direct contact with infected people	0.274 **	0.205 **	0.202 **
Depression		0.530 **	0.480 **
Resilience			−0.132 **
F	21.2 **	105.6 **	103.0 **
R^2^	0.096	0.368	0.383
ΔR^2^	0.096	0.272	0.015

* *p* < 0.05; ** *p* < 0.01.

**Table 3 ijerph-18-00636-t003:** Hierarchical linear regression analysis results (outcome variable: work-related burnout).

Variables	Step 1 (β)	Step 2 (β)	Step 3 (β)
Sex	0.198 **	0.117 *	0.109 *
Children ≤ 12y	0.096	0.130 **	0.125 **
Educational level			
*High school and below*	*Reference*		
*University degree*	0.305 *	0.185	0.198
*Postgraduate*	0.473 *	0.329 *	0.346 *
*Master’s*	0.391 **	0.313 **	0.327 **
*PhD*	0.284	0.192	0.234
Professional experience			
*Five years or less*	*Reference*		
*From 6 years to 15 years*	0.072	0.174 **	0.183 **
*More than 15 years*	0.004	0.100	0.103 *
Frontline working position	0.177 **	0.185 **	0.176 **
Diagnosed health problem	0.296 **	0.154 **	0.149 **
COVID-19 tested			
*Yes*	*Reference*		
*No, but I’d like to do it*	0.036	0.015	0.017
*No, I have no interest*	−0.128 *	−0.086	−0.088
Direct contact with infected people	0.272 **	0.199 **	0.196 **
Depression		0.522 **	0.476 **
Resilience			−0.118 **
F	10.4 **	68.8 **	67.7 **
R^2^	0.064	0.326	0.338
ΔR^2^	0.064	0.263	0.012

* *p* < 0.05; ** *p* < 0.01.

**Table 4 ijerph-18-00636-t004:** Hierarchical linear regression analysis results (outcome variable: client-related burnout).

Variables	Step 1 (β)	Step 2 (β)	Step 3 (β)
Professional experience			
*Five years or less*	*Reference*		
*From 6 years to 15 years*	0.114 *	0.187 **	0.194 **
*More than 15 years*	−0.081	−0.026	−0.026
Direct contact with infected people	0.153 **	0.111 *	0.106 *
Death of relative or friend during the pandemic period	−0.163	−0.189 *	−0.175 *
Depression		0.352 **	0.305 **
Resilience			−0.120 **
F	7.1 **	63.3 **	58.2 **
R^2^	0.014	0.137	0.149
ΔR^2^	0.014	0.122	0.012

* *p* < 0.05; ** *p* < 0.01.

## Data Availability

The exact data can be obtained from the corresponding author.

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
