# Peer review of "Burnout and Depression in Portuguese Healthcare Workers during the COVID-19 Pandemic—The Mediating Role of Psychological Resilience"

_ijerph, 2021, doi:10.3390/ijerph18020636_

Round 1

Reviewer 1 Report

No further comments.

Reviewer 2 Report

The article is well written. I have no objection to this version.

This manuscript is a resubmission of an earlier submission. The following is a list of the peer review reports and author responses from that submission.

Round 1

Reviewer 1 Report

Burnout in Healthcare workers (HCW) is not a specific phenomenon of the current situation. There is a wealth of literature regarding burnout in HCW. The prevalence of burnout in HCW outside the current COVID-19 situation has to be reviewed and to be compared against empirical findings in the actual situation. It has to be shown whether this current situation is really so specific in raising burnout in HCW.

The sampling has to be specified in much more detail. What professional organizations and health care institutions were involved ? How did the authors ensure that a representative sample of members of these organizations took part ?

The sample obviously is very heterogeneous. What exactly are „health technicians“? Are they really involved in patient care?

What is the rate of females working in the health care sector in Portugal? How does this rate correspond to the 83.6% in the current sample? The sample has to be examined for representativeness regarding HCW in Portugal, otherwise this is just a biased convenience sample.

Please explain: almost 70% state that they have a frontline position but 72.5% say that they do not have contact to infected people. Could this be a contradiction?

I do not capture this:

„COVID-19 Tested

Yes and, no but I'd like to do it“ => Contradiction ?

Line 187: The client related burnout was not high as it was <50. Please elaborate.

The descriptive statistics of the used questionnaires have to be presented in much more detail and compared to normative data.

Figure 1: The absolute effects are small and are due to the large sample size. Beta reduction from .53 to .48 and R square increase from .368 to .38 are statistically significant but not of practical importance.

Figure 2: The absolute effects are small and are due to the large sample size. Beta reduction from .522 to .476 and R square increase from .326 to .338 are statistically significant but not of practical importance.

Figure 3: The absolute effects are small and are due to the large sample size. Beta reduction from .352 to .305 and R square increase from .137 to .149 are statistically significant but not of practical importance.

Discussion

The comparisons of the prevalence of depression and other results in the current sample with other studies has to be more precise in that the samples in the other studies might have been different from their characteristics.

The Discussion and the Conclusions have to be rewritten and the association between resilience and burnout has to be attenuated as this is just a small effect which became significant just due to the large sample size.

Reviewer 2 Report

In this manuscript Serrão et al aim was to explore the mediating role of resilience in the relationship between depression and burnout in Portuguese healthcare workers during the COVID-19 pandemic.
The manuscript is well written and extends the current understanding of the impact of the COVID-19 pandemic on the mental health of healthcare workers.

Minor points,

The prevalence and incidence of COVID-19 in the area of study during the period of study should be discussed.

More abundant reference of previous works dealing with the subject of this study should be commented in the introduction or discussion: Wańkowicz, P.; Szylińska, A.; Rotter, I. Assessment of Mental Health Factors among Health Professionals Depending on Their Contact with COVID-19 Patients. Int. J. Environ. Res. Public Health 2020, 17, 5849.

Reviewer 3 Report

In the first place, I consider that the article is interesting and is in accordance with the current needs generated by Covid, where processes such as depression, anxiety and other types of disorders are being a clear consequence of this problem in health professionals.

Focusing on the article, it stands out that the abstract is well prepared and serves the purpose of introducing the reader to the study that is going to be read later.

Regarding the introduction, I miss more bibliographic references to provide greater consistency to this section. There is a lot of bibliography on the subject related to Psychological resilience as a protective factor, so a greater number of articles and authors should be referenced.

In relation to the methodology, it would be convenient to describe the data collection process a little more, detailing which were the inclusion and exclusion criteria when selecting the sample.

Finally, regarding the discussion, as in the introduction, a greater number of bibliographic references would be desirable to provide greater consistency to said section and thus ensure that the results obtained can be much more consistent.

Round 2

Reviewer 1 Report

I regret to say that I am still not satisfied with the responses of the authors to my comments.

Comment 2: Adding just 2 citations is not a thorough review as was demanded. Concrete numbers have to be extracted from these studies and compared to numbers from studies in the current situation.

Response to Comment 3 is not substantial and vague. Why are you mentioning professional organizations and health care institutions when you are not able to name them? This is unclear.

Comment 4: I disagree with category health technicians (e.g., physiotherapist, Pathological and cytological anatomy technician, oral hygienist, radiotherapy and radiology technician). These are completely heterogeneous health occupations which cannot be accumulated into one single category. A physiotherapist for example is by far not a „health technician“.

Comment 5: This is not a response to my objection. The global feminization rate in health is not of interest here but the rate in Portugal to ensure that this is not a biased convenience sample which I think it is.

Comment 7: I do not agree with this categorization. Having been tested and not having been tested but wanting to be tested are totally different categories from a methodological viewpoint. The intention-behavior gap always has to be considered in those who just state that they are willing to be tested/vaccinated etc..

Comment 11: The Conclusion still contains the recommendation to develop resilience training programs which is not supported by the small effects in the presented analyses. Therefore, the whole argumentation of the authors that resilience could be an adaptive personal resource for alleviating depression symptoms and mitigate burnout of HCW is not supported by the data but the authors are still following this argumentation which I cannot support from a methodological point of view.

Reviewer 3 Report

I have nothing more to comment. For my part it is accepted.